# How Regional High-Quality Co-Ordinated Development Influences Green Technology Choices: Evidence from 284 Cities in China

**Dameng Hu [1], Changbiao Zhong [1,2], Haoran Ge [3], Yawen Zou [4] and Chong Li [5,*]**

[1] School of Business, Ningbo University, Ningbo 315211, China; 2101010003@nbu.edu.cn (D.H.); zhongchangbiao@nbu.edu.cn (C.Z.)
[2] School of Economics, Guangzhou Business School, Guangzhou 511363, China
[3] School of Business, Zhejiang Wanli University, Ningbo 315100, China; gehaoran@zwu.edu.cn
[4] Ningbo Guozhan Engineering Consulting Company, Ningbo 315000, China; zyw0303@nbgodo.com
[5] School of Economics, Yunnan University of Finance and Economics, Kunming 650221, China
[*] Correspondence: lichong@ynufe.edu.cn

**Abstract:** High-quality development (HQD) is a fundamental requirement for current and future macroeconomic regulation in China. This study measured the high-quality co-ordinated development (HQCD) index of 284 cities in China from 2010 to 2019 using the entropy weighted TOPSIS method and coupled co-ordination model, and examined the impact of regional HQCD on enterprises' green technology choices by combining data from Chinese listed companies. The results show the following: (1) Regional HQCD significantly promotes enterprises' green technology choices, but does not substantially change the direction of their green technology progress. Specifically, co-ordinated regional economic–ecological system development promotes the enterprises' technological progress toward green practices. Moreover, co-ordinated urban development has a self-reinforcing effect on the preference for green technology choices. (2) Regional HQCD enhances the screening effect of enterprises on green technology by alleviating financial constraints and increasing the awareness of social responsibility. (3) Regional HQCD has a more pronounced promotional effect on green technologies in the categories of transportation; energy conservation; and administration, regulation, or design. Private enterprises and cities with a high-administrative rank responded to the green technology selection effect of regional HQCD. This study enriches the theory and literature on the influence of government policies on firm behavior, and also provides a reference for the international community.

**Keywords:** high quality development; green technology choices; the direction of green technology progress; self-reinforcing effect

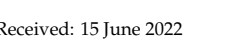



## 1. Introduction

Since the 21st century, China's economic level has achieved a leapfrog growth. From 2000 to 2021, China's GDP increased from USD 1.2 trillion to 17.7 trillion, jumping in ranking from sixth to second in the world. However, sloppy growth and the traditional approach of focusing only on GDP have also led to many problems in China's economic growth, such as an imbalance in resources and the environment, low economic efficiency, and widening regional disparity. These problems of unbalanced and insufficient development have severely constrained people's pursuit of a high-quality and diversified life [1]. In 2017, the Communist Party of China (CPC) first put forward the new important direction of high-quality development (HQD), clearly pointing out that China's economy had shifted from a high-speed growth stage to a high-quality development stage. Additionally, in 2020, it was again emphasized that "we must constantly improve our ability and level of implementing the new development concept and constructing a new development

pattern, so as to provide a fundamental guarantee for HQD". HQD has become the main theme of China's social development in the new era, and promoting HQD is an inevitable requirement to adapt to the change in the main contradiction in Chinese society and to build a modern socialist country [2].

At present, the high-quality development advocated by China aims to pursue interaction and co-ordination among five systems: the economic system, political system, cultural system, social system, and ecological system [3]. As China's economic development enters a new normal and its people yearn for a better life, it is imperative to speed up the transformation of the economic development pattern, focus on innovation and highlight green orientation. As an emerging technology designed to reduce energy consumption, reduce pollution, and improve ecology, green technology innovation has the dual advantages of realizing economic efficiency and environmental protection [4], and plays an important role in improving the green competitiveness of enterprises, alleviating pressure on resources and the environment and creating market demand [5]. In January 2019, China's Comprehensive Deepening Reform Commission adopted the "Guiding Opinions on Building a Market-Oriented Green Technology Innovation System", echoing the 19th CPC National Congress Report's request to "build a market-oriented green technology innovation system" and clearly stating that green technology innovation is an important driving force for green development and a key to promoting high-quality co-ordinated development (HQCD). Enterprises are innovative entities, but are also major sources of pollution. Whether the HQCD strategy implemented by the government can influence enterprises and which channels should be used to guide and screen enterprises' green technology practices are theoretical and practical issues that warrant further exploration.

In order to answer the above questions, this paper constructed an index system based on economic, political, cultural, social, and ecological systems, and accurately estimated the degree of HQCD of 284 cities in China from 2010 to 2019 by using the entropy weighted TOPSIS method and the coupling co-ordination degree model. Meanwhile, the CNRDS database and the "Green List of International Patent Classification" launched by the World Intellectual Property Organization (WIPO) are used to accurately identify the green patent information of Chinese listed companies. The empirical results show that the regional HQCD promotes the green technology choices of enterprises, but does not change the direction of green technology progress of enterprises. Specifically, the co-ordination of regional economy–ecology systems promotes enterprises' technological progress towards green practices. Moreover, regional co-ordinated development has a self-reinforcing effect on green technology choices. In addition, the regional HQCD can alleviate financial constraints and improve social responsibility, and thus enhance the screening effect of enterprises' green technology practices.

This paper has three main contributions: First, the design of the HQD indicator system in the previous literature mostly focuses on the level of provinces [6,7], with relatively few studies at the city level. Moreover, in the specific design process of the index system, there are three problems in most literature: first, the process index is incorrectly included, and the meanings of the result index and process index are confused; second, a similar index is added repeatedly; third, it only calculates the total score index of different systems, ignoring the interaction and co-ordination between systems. Therefore, this research strictly follows the principle of scientificity, operability and dynamism, on the basis of the economic, political, cultural, social, and ecological five systems, and designs a set of index systems containing 5 primary indicators and 24 secondary indicators, using the coupled co-ordination degree model to accurately calculate the HQCD of the 284 Chinese cities. Second, in contrast to the majority of literature studies on the pre-factors affecting HQD [8–11], this paper systematically studies the after-effects of HQCD for the first time by taking the green technology choices of enterprises as the entry point. Third, from the perspectives of the corporate financing environment and social responsibility, this paper conducts an in-depth investigation into the specific mechanism of regional HQCD affecting

corporate green technology screening, which makes relevant theories and the literature more complete and reasonable.

## 2. Literature Review and Theoretical Analysis

### 2.1. Economic Growth and HQD

HQD is a unique concept in China, while foreign studies focus on economic growth. In 1983, the Soviet economist Kamaeb first proposed the quality of economic growth, which focuses on the increase in production and living materials as well as the improvement of production efficiency and product quality. At present, there is still no unified definition of the concept of HQD. Kamaeb [12] argues that high-quality development embodies the "five development concepts" and should adhere to quality and effect first, then driving innovation, co-creation, and sharing. Zhao et al. [13] suggest grasping the connotation of HQD from three aspects: system balance, economic development, and people's livelihood orientation. In a broad sense, HQD is the co-ordinated development of economic construction, political construction, cultural construction, social construction, and ecological civilization construction [3], which embodies the new development concept and aims at meeting the needs of the people for a better life.

The evaluation and accounting of HQD is a complex and significant project. Many studies measure HQD from different dimensions, and some scholars use a single indicator of measurement, such as per capita real GDP [14], total factor productivity (TFP) [15,16], and green total factor productivity (GTFP) [9,17]. This measurement method is biased because the fundamental characteristic of HQD is multidimensional [18]. In recent years, constructing an evaluation index system to comprehensively measure HQD has become a hot topic. Scholars have explored the construction of the HQD index system from multiple perspectives, which has improved upon the limitations of previous studies. Chen et al. [6] only focuses on HQD in a narrow sense, and constructs an HQD indicator system for 30 provinces in China from three dimensions, namely, economic level, economic stability and economic sustainability. Yang et al. [7] incorporate environmental factors into the study from the aspects of the ecological environment, economic structure, and economic efficiency. Miao and Feng [19] take a different approach to constructing HQD indicators from micro-, medium, and macro-perspectives. However, more studies build indicator systems based on the five development concepts of innovation, co-ordination, green, openness and sharing [8,10,20–23].

Existing studies on the antecedents of HQD have mostly focused on science and technology innovation [24–26], FDI [8,27], environmental regulation [9,27], the opening of high-speed rail [28], the digital economy [10,29], human capital [11] and industrial structure [6,30,31]. Additionally, the literature dealing with the posterior impact of HQD is still in the exploratory stage.

### 2.2. Innovation and Green Technology Options

In the era of the knowledge-based economy, technological innovation is particularly important for the development of enterprises and national economies, as it is a fundamental means to improve product structure, increase the added value of products, and enhance the competitiveness of enterprises, as well as a major motivation for the advanced industrial structure and a fundamental source of economic growth [32]. However, according to the theory of biased technological progress [33], innovation is directional, and firms can engage in both "clean" and "polluting" innovations. This means that some of the positive effects resulting from an increase in the number of innovations may not have an energy–environmental effect either. For this reason, Rhodes and Wield [34] proposed the concept of green innovation, which can be defined as a generic term for processes, technologies, or products that reduce energy and raw material consumption and environmental pollution and promote sustainable development [35]. Compared with general corporate innovation, green innovation can help companies achieve both economic and

environmental benefits, and is an important way to mitigate the negative environmental impacts of economic activities [4].

The study of factors influencing firms' green innovation behavior has been a cutting-edge academic topic, and established studies have resulted in rich discussions. Some scholars have discussed the factors within firms. Zhong and Yang [36], based on the theory of planned behavior, found that state-owned enterprises have a stronger willingness to innovate with green technology, and thus, their green innovation is deeper compared to private firms [37]. Zhang and Wang [38] studied the role of two types of management relationships in green innovation and confirmed that corporate business relationships had a positive effect on green innovation, while corporate political relations had an inverted U-shaped effect on green innovation. Currently, there is an increasing diversity of factors that have been mined regarding intra-firm factors, including digital development [39–41], executive heterogeneity [42–44], corporate social responsibility [45] and corporate merger behavior [46]. However, as research has progressed, a large body of literature has begun to turn to the study of the effects of green technology innovation on factors external to the firm. According to the Porter hypothesis [47], flexible and reasonable environmental regulations do not pose much of a barrier to business operations and are effective in stimulating firms' willingness to innovate and improve their innovation performance. Subsequently, a large body of literature has focused on the differential effects of green innovation from different types of environmental regulations, including command and control [48–51], market driven [52–54], and voluntary participation [54–56]. Additionally, the implementation of many government policies also affects firms' green behavior; Zhang et al. [57] explored the effect of green credit policies on the green innovation of highly polluting firms in China and showed that green credit policies improved overall and incremental green innovation, but hindered the fundamental green innovation of highly polluting firms [58]. Lu and Wang [59] investigated the national Five-Year Plan for environmental protection and found that the policy induced green innovation at both the regional level and the industrial level. Du et al. [60] used China's pilot policy on carbon emissions trading as a quasi-natural experiment and confirmed that the implementation of the policy significantly promoted green innovation among firms in the pilot region, but had a dampening effect on its neighboring regions.

*2.3. Research Hypothesis*

China's sloppy economic growth over the years has led to a series of problems, such as the serious waste of natural resources, economic inefficiency, and damage to the ecological environment [61]. According to the 2021 China Ecological Environment Status Bulletin, if the effects of sand and dust are not deducted, the ambient air quality in a total of 146 cities out of 339 cities in China exceeded the standard, accounting for 43.1%, and the average ratio of days exceeding the standard was 12.5%. The deterioration of the ecological environment has become the primary factor hindering the overall development of cities. Additionally, the current strategy of HQD advocated by China is a product of the co-ordinated development of the economic system, political system, cultural system, social system, and ecosystem in five parts [3]. Local governments actively respond to the call of the central government on the strategy of HQD, that it is necessary to prioritize urban ecology. As enterprises in various industries in cities are major sources of pollution [62], determining how to improve the economic efficiency of enterprises while considering the environmental and social benefits of enterprises has undoubtedly become the focus of local governments.

Compared to traditional innovation, green technology innovation places more emphasis on non-pollution, low energy consumption, recyclability, and cleanliness [63]; therefore, under the regulatory requirements of local governments to promote high-quality interactive and coordinated development in various fields, green technology innovation is gradually becoming an important means for enterprises to pursue the unification of economic, social, and environmental benefits [4]. Additionally, the variation in the HQCD state from non-existence to existence and from weakness to strength within and between regions gives

enterprises the space to change from accepting and adapting to making changes completely. In other words, regions with a HQCD status running for a long time and with a high level of co-ordination have a stronger possibility, as well as extent, of enterprises catering to the government and making corresponding green technology choices. Therefore, the following hypothesis is proposed in this paper.

**Hypothesis 1 (H1).** *Regional HQCD has a screening effect on enterprises' green technology choices, and this feature shows a tendency to strengthen continuously with an increase in the level of regional co-ordinationl.*

Enterprises need a large amount of resources and capital to support R&D innovation, and for green technology R&D and innovation activities, due to their greater technical difficulty, longer payback periods and externalities with private costs that are significantly greater than social costs. Enterprises need strong support from outside society and the government, as well as more diverse resources and a certain market position, which is more inseparable from the continuous investment of R&D capital [64]. Debt financing and equity financing are important sources of exogenous financing for enterprises to obtain sufficient funds [65]. In China, due to the high threshold of equity financing and the long application cycle, debt financing from banks is still an important way for enterprises to raise funds when liquid equity capital is scarce [66]. In the strategic context of HQCD, local governments must spare no effort to make up for shortcomings, promote the reform of the financial system, optimize the allocation of financial resources, and provide a more relaxed financing environment for enterprises, thus emphasizing the vitality of SMEs and promoting the overall improvement of their green technology level. Therefore, this paper proposes the following hypothesis.

**Hypothesis 2 (H2).** *Regional HQCD will help firms in green technology screening by reducing financial constraints.*

According to stakeholder theory, the stakeholders of a firm are divided into market stakeholders and non-market stakeholders. Non-market stakeholders include the government, social groups, and the media, who have no clear material interests in the enterprise and are objects that are not directly involved in the production and operation of the enterprise [67]. Corporate social responsibility (CSR), on the other hand, refers to the enterprise's responsibility to employees, shareholders, and even the government and the public while creating profits. From the proposal, implementation, promotion, and deepening of regional HQD strategies, a series of initiatives by local governments will also have an impact on CSR. As an important non-market stakeholder of enterprises, the HQD pattern promoted by the government is undoubtedly conveying a policy orientation to enterprises. By actively fulfilling social responsibility and maintaining good relationships with non-market stakeholders, enterprises avoid government penalties and media opinion pressure, etc., which enhances their image in the minds of the public and government, and their social status may be improved [68]. Additionally, the active fulfillment of social responsibility by enterprises directly affects the behavior of market stakeholders, among which investors have a significant positive response to socially responsible behavior. When making investment decisions, investors will consider not only the return of corporate stocks, but also examine the fulfillment of corporate social responsibility in a comprehensive manner. Companies with a high degree of social responsibility fulfillment are more likely to attract high-quality investors, receive more financial support, acquire more resources for green technology research and development, and become more successful [69]. Therefore, the following hypothesis is proposed in this paper.

**Hypothesis 3 (H3).** *Regional HQCD will promote corporate green technology choices through increased CSR.*

## 3. Research Design

### 3.1. Sample Selection and Data Sources

Considering the availability of some indicators and data integrity, in terms of the time dimension, this study selected 10-year data from 2010 to 2019. At the city level, this paper selected 284 prefecture-level cities and above in China as the research objects, and studied the cities with serious indicators missing or administrative level changes that were not included in the sample period,, such as Lhasa, Turpan, Chaohu, Tongren, Bijie, Sansha, and Haidong. The city-level data were obtained from the China Urban Statistical Yearbook, the China Statistical Yearbook, and the data on the relationship between the government and businesses in cities released by the National Institute of Strategy and Development, Renmin University of China, while the data of PM2.5 and SO2 mass concentration in cities were collected and summarized using NASA's M2TMNXAER_5.12.4 satellite data. Some of the missing values were queried and completed using statistical yearbooks of provinces, the EPS database, and the China Economic Network database. At the enterprise level, this paper selected Chinese A-share listed companies in Shanghai and Shenzhen stock markets as research samples, and the data were derived from the CSMAR and CNRDS databases. The green patent information of enterprises involved was separately segmented and extracted item by item from the CIRD sub-database in the CNRDS database by using the "Green List of International Patent Classification" launched by WIPO, and, finally, classified and summarized. The CSR score data were obtained by crawling the social responsibility data of listed companies on the Hexun website. According to the previous relevant literature, the samples with abnormal financial status or other abnormal conditions (namely ST, *ST and S*ST enterprises), the samples with statistical or calculation errors, and the enterprises with a duration of 5 years or more were deleted, and finally, the samples of 2422 listed companies were obtained.

### 3.2. Variables Design

#### 3.2.1. Independent Variable Measurement and Result Presentation

The essence of China's HQD is efficient, fair, green and sustainable development aimed at achieving a better quality of life, and it is a co-ordinated development of economic, political, cultural, social and ecological progress [3]. Based on this, this paper constructs a comprehensive evaluation index system of HQD at the city level from the five systems of economy, politics, culture, society, and ecology. See Table 1 for the selection and calculation methods of specific evaluation indicators.

**Table 1.** Evaluation index system for regional HQD.

| System | Evaluation Index | Calculation Methods | Index Attribute | Weight |
|---|---|---|---|---|
| Economy | GDP per capita | GDP/population (RMB/person) | + | 0.5013 |
| | Total factor productivity | Calculated using the latest SFA method, where output is set as real GDP and input factors are the number of employees and fixed assets (−) | + | 0.1944 |
| | Average wage of employees | Annual total salary of employees/annual average number of employees (RMB/person) | + | 0.2252 |
| | Economic efficiency of water use | Annual total water consumption/real GDP (tons/10 thousand RMB) | − | 0.0515 |
| | Economic efficiency of electricity use | Annual electricity consumption/real GDP (kWh/10 thousand RMB) | − | 0.0276 |

**Table 1.** *Cont.*

| System | Evaluation Index | Calculation Methods | Index Attribute | Weight |
|---|---|---|---|---|
| Politics | The size of government finances | Local general public budget expenditure/GDP (−) | + | 0.3814 |
| | Fiscal decentralization | Local general public budget revenue/local general public budget expenditure (−) | + | 0.2753 |
| | Proportion of public sector staff | Number of employees in public administration and social organizations/total number of employees in the tertiary industry (%) | + | 0.1456 |
| | Government transparency | The National Institute of Development and Strategy of Renmin University of China has created an evaluation system for the health index of government-business relations, which measures the health index of government-business relations in cities in China, including the first-level indicators, government transparency and government integrity (−) | + | 0.1300 |
| | Government integrity | Same as above (−) | − | 0.0677 |
| Culture | Number of public books per capita | Public library book holdings/population (volume/100 people) | + | 0.4146 |
| | Number of full-time teachers | Total number of full-time teachers in regular higher schools, regular secondary schools and regular primary schools/population (−) | + | 0.1163 |
| | Number of college students | Number of students in regular colleges and universities/population (−) | + | 0.4328 |
| | Proportion of employees in cultural-related industries | Number of employees in culture, education, sports and entertainment/total number of employees in the tertiary industry (%) | + | 0.0363 |
| Society | Internet penetration rate | Number of households connected to the Internet/total number of households (%) | + | 0.3242 |
| | Urban unemployment | Number of registered unemployed/total labor force (%) | - | 0.0014 |
| | Number of doctors | Number of practicing and assistant physicians/population (−) | + | 0.1974 |
| | Use area of road | Actual road area/population ($m^2$/person) | + | 0.2158 |
| | Number of buses and trams | Number of buses and trams in operation/population (vehicle/10 thousand people) | + | 0.2611 |
| Ecology | PM2.5 concentration | Using NASA's M2TMNXAER_5.12.4 satellite data, the raster data in China is cut and summarized by city to obtain PM2.5 mass concentration ($\mu g/m^3$) | - | 0.1297 |
| | $SO_2$ concentration | Using NASA's M2TMNXAER_5.12.4 satellite data, the raster data in China is cut and summarized by city to obtain $SO_2$ mass concentration ($\mu g/m^3$) | - | 0.1592 |
| | Harmless treatment rate of domestic waste | Quantity of harmless disposal of domestic waste/production (%) | + | 0.0694 |
| | Industrial solid waste utilization | Effective utilization of industrial solid waste/production (%) | + | 0.1158 |
| | Green area | The total area of various green spaces/population ($m^2$/person) | + | 0.5260 |

The co-ordination degree of regional HQD was the independent variable of this paper. Before the calculation of the co-ordination degree, this study used the entropy weighted TOPSIS method to assign weight to the relevant indicators [70] and obtain the comprehensive index score of the city based on five systems in the sample period. Then, the coupling co-ordination degree model was used to calculate the co-ordination degree of the five systems [71]. The specific calculation process is shown in Appendix A. Meanwhile, the paper provides a detailed cross-period and cross-regional description of the final measurement results; see Appendix B.

### 3.2.2. Dependent Variable

The dependent variables are green technology (GT) and green technology progress direction (GTPD). The advantage of patents as a measure of innovation is that they are easy to obtain and can be broken down into different technology areas. Additionally, one of the main outputs of the green innovation process is the green patent. In this paper, two indicators, the number of green patents applied by listed companies in the current year and the percentage of green patents applied, were selected to indicate the level of green technology and the green technologyl progress of enterprises, respectively. Compared with the patent grant, the patent application better represents the achievements in the technological innovation of enterprises in the year, while patents are often granted only 1–3 years after application [72], and the patent grant is subject to many instabilities due to factors such as testing, annual fee payments, and market environment [73]. In addition, compared with the number of green patents alone, the percentage of green patents reflects the relative importance of green patents and represents the direction of green technological progress of enterprises, while effectively eliminating other unobservable factors that can have an impact on innovation [74,75].

### 3.2.3. Control Variables and Mechanism Variables

The control variables contain variables such as firm size (Size), age (Age), capital intensity (CapiInten), main profit margin (MainProfMarg), and social wealth creativity (SociWealCrea). In this paper, the logarithm of the number of employees in a firm is used to indicate firm size, which is generally considered to be more likely to lead to economies of scale, more conducive to controlling resources and producing better employees [76], and thus positively associated with innovation performance. Using the age of the firm listed to indicate the age of the firm, one view suggests that the firm's sense of innovation is positively related to its time of establishment [77], but another view, based on the theory of organizational inertia, suggests that the firm's age is an important barrier to innovation [78]. Whereas firms with more fixed assets will seek innovation and environmental behavior more actively than firms with fewer fixed assets, because the former have a greater risk of failure [79], this paper uses the logarithm of fixed capital per capita to express capital intensity. The ratio of operating profit is used to maintain business income to express the main profit margin. TobinQ is an important and widely accepted measure of firm performance [80], which represents the relationship between the market value of a firm and the replacement value of its assets [81], and is used to represent social wealth creativity. The descriptive statistics of related variables are shown in Table 2.

In addition, to test hypotheses 2 and 3, two mechanism variables were introduced in this paper, which are the financing constraint of the firm (FinanConstra) and the social responsibility of the firm (CSR). The financing constraint variable of the firm is obtained by using the firm's interest expense/average total debt [82], and its larger value implies that the firm is facing a greater degree of financing constraint. While the social responsibility score of enterprises was obtained by crawling Hexun.com data (http://stockdata.stock.hexun.com/zrbg/, accessed on 12 July 2022) on listed companies' responsibility reports, the professional evaluation system of listed companies' social responsibility reports examines five items comprising shareholder responsibility, employee responsibility, supplier, customer and consumer rights responsibility, environmental responsibility, and social responsibility, and

this paper used the total social responsibility score of listed companies aggregating the five items as corporate social responsibility.

**Table 2.** Summary statistics.

|  | N | Mean | SD | Min | Max |
|---|---|---|---|---|---|
| GT | 21,600 | 1.124 | 4.252 | 0.00 | 32.00 |
| GTPD | 21,600 | 0.045 | 0.145 | 0.00 | 1.00 |
| Co-ordination | 21,600 | 0.526 | 0.120 | 0.24 | 0.79 |
| Co-ordination_EE | 21,600 | 0.677 | 0.154 | 0.37 | 0.95 |
| Co-ordination_PCS | 21,600 | 0.446 | 0.107 | 0.18 | 0.70 |
| Size | 21,599 | 7.672 | 1.355 | 4.03 | 13.22 |
| Age | 21,600 | 10.558 | 7.107 | 0.00 | 29.00 |
| CapiInten | 21,581 | 12.487 | 1.214 | 4.13 | 19.53 |
| MainProfMarg | 21,579 | 0.082 | 0.207 | −1.06 | 0.66 |
| SociWealCrea | 21,600 | 2.291 | 1.957 | 0.18 | 11.81 |
| FinanConstra | 19,274 | 0.021 | 0.015 | 0.00 | 0.07 |
| CSR | 21,553 | 24.991 | 17.001 | −18.45 | 90.87 |
| HumaCapi | 21,600 | 0.048 | 0.039 | 0.00 | 0.24 |
| FinanScal | 21,600 | 0.156 | 0.056 | 0.07 | 0.62 |
| FDI | 14,650 | 0.208 | 0.145 | 0.00 | 0.51 |

### 3.3. Model Setting

This paper measured the HQCD of 284 cities in China from 2010 to 2019 at the city level, so as to explore the impact of regional HQCD on the bias of firms' green technology choices. The econometric model is set as follows.

$$y_{ijt} = \beta_0 + \beta_1 \text{Coordination}_{it} + \beta_j \text{Controls}_{ijt} + \text{Year FE} + \text{Region FE} + (\text{Year} * \text{Region})\text{FE} + \varepsilon_{ijt} \qquad (1)$$

where $y_{ijt}$ denotes the green technology (GT) and green technology progress direction (GTPD) of firm j in city i in year t, respectively. $\text{Coordination}_{it}$ denotes the co-ordination of HQD in city i in year t, and $\text{Controls}_{ijt}$ denotes a series of control variables. The fixed effects include time fixed effects, region fixed effects, and time × region fixed effects. In addition, considering that the independent variable dimension is regional and the dependent variable is firm dimension, and the standard errors of the panel data are underestimated due to the autocorrelation of the nuisance terms in both individual and temporal dimensions, the standard errors of the regressions were adjusted by clustering in the regional dimension, following the design of Moshiriana et al. [83].

## 4. Empirical Results

### 4.1. Baseline Results

Table 3 presents the regression results of the impact of regional HQCD on enterprises' green technology practices and the direction of green technology progress. Model 1 and Model 2 show the regression results of enterprises' green technology practices, and it can be observed that the coefficient of co-ordination in Model 1 is positive and significant, while the coefficient of Model 2 is still positive and significant after controlling for the time fixed effect, region fixed effect and time × region fixed effect. On the contrary, Model 3 and Model 4 show the regression results on the direction of green technology progress of enterprises, and it is obvious that the coefficient of co-ordination is positive and insignificant, regardless of the three types of fixed effects, which indicates that the evolution of regional HQCD does not change the direction of green technology progress of local enterprises and that HQCD not only promotes green technology, but also may, for some reasons, partially promote non-green technology at the same time, leading to the fact that regional HQCD does not substantially affect the direction of green technology progress of enterprises.

**Table 3.** The impact of HQCD on green technology choice.

|  | GT | | GTPD | |
| --- | --- | --- | --- | --- |
|  | **(1)** | **(2)** | **(3)** | **(4)** |
| Co-ordination | 5.473 *** | 2.421 ** | 0.041 | 0.022 |
|  | (1.460) | (1.138) | (0.029) | (0.020) |
| Size | 1.860 *** | 1.828 *** | 0.009 *** | 0.008 *** |
|  | (0.519) | (0.522) | (0.001) | (0.001) |
| Age | −0.040 | −0.019 | −0.002 *** | −0.002 *** |
|  | (0.026) | (0.025) | (0.000) | (0.000) |
| CapiInten | 0.375 | 0.403 | 0.003 | 0.003 |
|  | (0.246) | (0.262) | (0.002) | (0.002) |
| MainProfMarg | −0.004 ** | −0.003 ** | −0.000 | −0.000 ** |
|  | (0.002) | (0.002) | (0.000) | (0.000) |
| SociWealCrea | 0.010 | 0.009 | −0.000 | 0.000 |
|  | (0.009) | (0.009) | (0.000) | (0.000) |
| Year | No | Yes | No | Yes |
| Region | No | Yes | No | Yes |
| Year-Region | No | Yes | No | Yes |
| Constant | −19.746 *** | −18.469 ** | −0.060 | −0.040 |
|  | (6.892) | (6.829) | (0.036) | (0.034) |
| N | 21,566 | 21,566 | 21,566 | 21,566 |
| Adj. $R^2$ | 0.041 | 0.033 | 0.016 | 0.017 |

Note: figures in () are standard error; *** and ** indicate significance at the 1% and 5% levels, respectively.

In order to further investigate the reasons that the regional HQCD has not been implemented to change the direction of enterprises' green technological progress, this paper split the "five-in-one" HQCD into two parts: one is the co-ordinated development of the regional economic–ecological system, which is more closely related to both green practices and innovation, and the other is the co-ordinated development of the regional political–cultural–social system. The regression results are shown in Table 4. In Table 4, the coefficients of Co-ordination_EE in both Model 1 and Model 2 are positive and significant, indicating that the co-ordinated development of the regional economy–ecosystem not only has a screening effect on the green technology choices of enterprises, but also promotes technological progress towards green practices. On the contrary, the coefficients of Co-ordination_PCS are positively significant in Model 3, but positively insignificant in Model 4, and the results are consistent with the corresponding results in Table 3. In summary, the "five-in-one" regional HQCD may need to consider too many disturbing factors in the implementation process, such as the urban construction of people's lives and the regional development of high technology, which have to include many non-green factors, resulting in an insubstantial effect on the direction of green technology progress. To the contrary, the focus is solely placed on the co-ordinated development between the two systems of economy and ecology, which not only has a screening effect on green technology, but will also influence the direction of technological progress.

**Table 4.** The impact of economy–ecosystem co-ordination on GTPD.

|  | Economy–Ecology | | Politics–Culture–Society | |
| --- | --- | --- | --- | --- |
|  | **GT** | **GTPD** | **GT** | **GTPD** |
|  | **(1)** | **(2)** | **(3)** | **(4)** |
| Co-ordination_EE | 1.724 * | 0.033 ** |  |  |
|  | (0.879) | (0.015) |  |  |
| Co-ordination_PCS |  |  | 2.601 * | 0.014 |
|  |  |  | (1.342) | (0.022) |
| Size | 1.827 *** | 0.008 *** | 1.829 *** | 0.008 *** |
|  | (0.522) | (0.001) | (0.522) | (0.001) |

**Table 4.** *Cont.*

| | Economy–Ecology | | Politics–Culture–Society | |
|---|---|---|---|---|
| | GT | GTPD | GT | GTPD |
| | (1) | (2) | (3) | (4) |
| Age | −0.018 | −0.002 *** | −0.019 | −0.002 *** |
| | (0.024) | (0.000) | (0.025) | (0.000) |
| CapiInten | 0.398 | 0.003 | 0.404 | 0.003 |
| | (0.262) | (0.002) | (0.262) | (0.002) |
| MainProfMarg | −0.003 ** | −0.000 ** | −0.003 ** | −0.000 ** |
| | (0.002) | (0.000) | (0.002) | (0.000) |
| SociWealCrea | 0.009 | 0.000 | 0.010 | 0.000 |
| | (0.009) | (0.000) | (0.009) | (0.000) |
| Year | Yes | Yes | Yes | Yes |
| Region | Yes | Yes | Yes | Yes |
| Year-Region | Yes | Yes | Yes | Yes |
| Constant | −18.292 ** | −0.051 | −18.376 ** | −0.034 |
| | (6.892) | (0.034) | (6.803) | (0.034) |
| N | 21,566 | 21,566 | 21,566 | 21,566 |
| Adj. $R^2$ | 0.033 | 0.017 | 0.033 | 0.017 |

Note: figures in () are standard error; ***, ** and * indicate significance at the 1%, 5%, and 10% levels, respectively.

This paper further compares the differences in green technology choices among different levels of HQCD, and divides cities into three categories, namely, level 1, level 2, and level 3, using the 25% and 75% quartiles of co-ordination as thresholds, respectively, to construct interaction terms between city-level dummy variables and coordinated development, so as to explore the differences in green technology choices among different levels of co-ordinated development. Table 5 Model 1 to Model 3 show the regression results of HQCD on green technology for first-tier, second-tier, and third-tier cities, respectively. It can be seen that the regression coefficients of second-tier and third-tier cities are not significant, while the regression coefficients of first-tier cities are positively significant, indicating that as the level of HQCD of cities increases, its effect on the green technology choices of enterprises gradually increases. Models 4 to 6 show the regression results of the economic–ecological system co-ordination development on the direction of green technology progress for first-tier, second-tier, and third-tier cities, respectively. It can be observed that the regression coefficient is positively significant only for first-tier cities, indicating that as the level of urban economic–ecological system co-ordination increases, it will accelerate the transformation of enterprise technology toward green practices. This further verifies hypothesis 1 that co-ordinated urban development has a self-reinforcing effect on green technology selection preferences.

**Table 5.** Self-reinforcing effect test of co-ordinated development on green technology choice.

| | GT | | | | GTPD | |
|---|---|---|---|---|---|---|
| | (1) | (2) | (3) | (4) | (5) | (6) |
| Level1*Co-ordination | 0.805 *** | | | | | |
| | (0.266) | | | | | |
| Level2*Co-ordination | | 0.390 | | | | |
| | | (0.499) | | | | |
| Level3*Co-ordination | | | −1.205 | | | |
| | | | (0.743) | | | |
| Level1*Co-ordination_EE | | | | 0.018 ** | | |
| | | | | (0.007) | | |
| Level2*Co-ordination_EE | | | | | 0.002 | |
| | | | | | (0.010) | |

**Table 5.** *Cont.*

| | GT | | | | GTPD | |
|---|---|---|---|---|---|---|
| | **(1)** | **(2)** | **(3)** | **(4)** | **(5)** | **(6)** |
| Level3*Co-ordination_EE | | | | | | −0.009 * |
| | | | | | | (0.005) |
| Control Variables | Yes | Yes | Yes | Yes | Yes | Yes |
| Year | Yes | Yes | Yes | Yes | Yes | Yes |
| Region | Yes | Yes | Yes | Yes | Yes | Yes |
| Year-Region | Yes | Yes | Yes | Yes | Yes | Yes |
| Constant | −17.257 ** | −17.137 ** | −17.001 ** | −0.033 | −0.027 | −0.023 |
| | (6.748) | (6.694) | (6.683) | (0.033) | (0.033) | (0.033) |
| N | 21,566 | 21,566 | 21,566 | 21,566 | 21,566 | 21,566 |
| Adj. $R^2$ | 0.033 | 0.032 | 0.033 | 0.018 | 0.017 | 0.017 |

Note: figures in () are standard error; ***, ** and * indicate significance at the 1%, 5%, and 10% levels, respectively.

### 4.2. Robustness Tests

To improve the reliability of the correlation results, four robustness tests were adopted in this paper to obtain the main results. The first one replaces the dependent and independent variables. The above baseline regression uses the green patent application indicator, but in the Chinese scenario, patent application activities may be full of false patents and unqualified patents, forming an "innovation illusion" [84], while patent acquisition can truly reflect the innovation ability of an enterprise. Therefore, this paper reconstructed two indicators of green patent acquisition and the green patent acquisition ratio to indicate the direction of green technology and green technology progress. From Model 1 and Model 2 in Table 6, it can be observed that the coefficients of both Co-ordination and Co-ordination_EE are positively significant after replacing the variables, which means that the main conclusion of this paper holds.

**Table 6.** Robustness testing of substitution variables and addition of control variables.

| | Change the Measurement Method of Independent and Dependent Variables | | Add City-Level Control Variables | |
|---|---|---|---|---|
| | **GT** | **GTPD** | **GT** | **GTPD** |
| | **(1)** | **(2)** | **(3)** | **(4)** |
| Co-ordination | 1.689 ** | | 3.562 ** | |
| | (0.783) | | (1.619) | |
| Co-ordination_EE | | 0.064 * | | 0.035 ** |
| | | (0.036) | | (0.015) |
| HumaCapi | | | 13.311 | 0.075 * |
| | | | (10.404) | (0.045) |
| FinanScal | | | 11.720 * | 0.101 ** |
| | | | (6.856) | (0.041) |
| FDI | | | −0.155 | −0.027 |
| | | | (2.434) | (0.019) |
| Control Variables | Yes | Yes | Yes | Yes |
| Year | Yes | Yes | Yes | Yes |
| Region | Yes | Yes | Yes | Yes |
| Year-Region | Yes | Yes | Yes | Yes |
| Constant | −12.171 *** | −0.068 ** | −24.764 *** | −0.060 *** |
| | (4.240) | (0.028) | (4.247) | (0.022) |
| N | 21,566 | 21,566 | 14,617 | 14,617 |
| Adj. $R^2$ | 0.022 | 0.036 | 0.052 | 0.025 |

Note: figures in () are standard error; ***, ** and * indicate significance at the 1%, 5%, and 10% levels, respectively.

Second, as certain variables at the city level may have an impact on both co-ordination and green technology selection preferences, this paper included city-level human capital

level (HumaCapi), fiscal size (FinanScal), and foreign investment share (FDI) in the regressions to control for possible endogeneity issues. Due to the limitation of some indicators, the sample years are 2010–2016, and the regression results are presented in Model 3 and Model 4 in Table 6, where the coefficients of both co-ordination and Co-ordination_EE are positively significant after controlling for city-level related factors, and the conclusions of this paper still hold.

Third, since the four municipalities directly under the central government of Beijing, Shanghai, Tianjin, and Chongqing belong to the provincial administrative level, and the functions and positioning undertaken by ordinary prefecture-level cities are obviously different, which may sway the green innovation effect of co-ordinated development, Model 1 and Model 2 in Table 7 present the regression results excluding the four municipalities directly under the central government. The results show that the baseline regression findings are still robust.

**Table 7.** Robustness test for excluding some samples and controlling for other policies.

| | Remove Samples from Municipalities | | Exclude Other Environmental Policies | |
|---|---|---|---|---|
| | GT | GTPD | GT | GTPD |
| | (1) | (2) | (3) | (4) |
| Co-ordination | 1.953 * | | 1.310 * | |
| | (1.023) | | (0.761) | |
| Co-ordination_EE | | 0.031 ** | | 0.044 *** |
| | | (0.015) | | (0.010) |
| Newly revised Ambient Air Quality Standards policy | | | 0.377 | −0.010 ** |
| | | | (0.249) | (0.004) |
| Special emission limit policy for air pollutants | | | 0.032 | 0.001 |
| | | | (0.179) | (0.004) |
| New environmental protection law policy | | | −0.561 * | 0.001 |
| | | | (0.302) | (0.003) |
| Control Variables | Yes | Yes | Yes | Yes |
| Year | Yes | Yes | Yes | Yes |
| Region | Yes | Yes | Yes | Yes |
| Year-Region | Yes | Yes | Yes | Yes |
| Constant | −11.707 ** | −0.030 * | −18.303 *** | −0.052 *** |
| | (4.746) | (0.017) | (2.507) | (0.017) |
| N | 17,023 | 17,023 | 21,566 | 21,566 |
| Adj. $R^2$ | 0.017 | 0.009 | 0.045 | 0.024 |

Note: figures in () are standard error; ***, ** and * indicate significance at the 1%, 5%, and 10% levels, respectively.

Fourth, the preferences of enterprises' green technology choices are inevitably influenced by relevant environmental policies, for which three large national environmental policies are collected and compiled in this paper, including the newly revised Ambient Air Quality Standards implemented in 74 cities in 2012, the special emission limit value policy for air pollutants emphasized in the national key regional air pollution prevention and control plan implemented in 2013, and the new environmental protection law implemented in 2015 mainly for heavily polluting enterprises. This is completed by including relevant policy dummy variables and cross terms of their time trends in the regressions as a way to control for the impact of relevant environmental policies on the use of green technologies by enterprises. The results are shown in Table 7, Model 3, and Model 4, where the coefficients of co-ordination and Co-ordination_EE are still positively significant after controlling for the possible green effects of relevant environmental policies, indicating that the conclusions of this paper are robust.

### 4.3. Mechanism Analysis

The results of both benchmark regressions and robustness tests confirm that regional high-quality coordinated development drives firms' green technology selection behavior, while this section tests hypotheses 2 and 3 from two perspectives: firms' financial constraints and firms' social responsibility, based on the theoretical analysis in the previous section.

Model 1 and Model 2 in Table 8 show the regression results of regional HQCD on corporate financial constraints (FinanConstra) and corporate social responsibility (CSR), respectively. The coefficients of co-ordination are negatively significant and positively significant, indicating that in the process of regional HQCD, the government optimized the market financing environment and the enterprises catered to the government policy orientation, alleviating the financial constraints of enterprises and improving awareness of social responsibility, which is consistent with the above theoretical analysis. Model 3 and Model 4 show the regression results of corporate financial constraints and social responsibility on green technology selection. The results show that corporate financial constraints significantly inhibit green technology selection, while corporate social responsibility consciousness significantly promotes the screening effect of green technology. From Model 1 to Model 4, regional HQCD enhances the screening effect of green technology by alleviating the financing environment of enterprises and improving corporate social responsibility, thus verifying hypotheses 2 and 3.

**Table 8.** Mechanism test of HQCD on screening effect of green technology.

| | FinanConstra | CSR | GT | GT |
|---|---|---|---|---|
| | (1) | (2) | (3) | (4) |
| Co-ordination | −0.020 *** | 7.170 *** | 2.601 ** | 2.479 ** |
| | (0.003) | (1.330) | (1.187) | (1.094) |
| FinanConstra | | | −6.024 ** | |
| | | | (2.913) | |
| CSR | | | | 0.001 *** |
| | | | | (0.000) |
| Control Variables | Yes | Yes | Yes | Yes |
| Year | Yes | Yes | Yes | Yes |
| Region | Yes | Yes | Yes | Yes |
| Year-Region | Yes | Yes | Yes | Yes |
| Constant | 0.013 *** | −9.854 *** | −20.352 *** | −18.518 *** |
| | (0.003) | (1.509) | (1.373) | (1.241) |
| N | 19,261 | 21,519 | 19,261 | 21,519 |
| Adj. $R^2$ | 0.046 | 0.190 | 0.033 | 0.034 |

Note: figures in () are standard error; *** and ** indicate significance at the 1% and 5%, levels, respectively.

### 4.4. Heterogeneity Analysis

#### 4.4.1. Patent Category Heterogeneity

According to the Green List of International Patent Classification provided by WIPO, green patents are divided into seven categories: alternative energy production, transportation, energy conservation, waste management, agriculture and forestry, administrative regulations and design, and nuclear power. The number of agriculture, forestry and nuclear power patents in this study sample is too small to be representative, so the remaining five patent categories are selected to investigate the difference in HQCD in the selection of different types of green technology. Table 9 reports the corresponding regression results. The results show that regional HQCD mainly promotes enterprises' research and development of green technologies in transportation, energy conservation, administrative regulation and design, but has no significant incentive effect on green technologies in alternative energy production and waste management. A possible reason for this is that, for a long time, the Chinese government has advocated for the concept of sustainable development already entrenched. Areas will more or less choose traditional concepts, which makes it relatively

easy to make achievements in technology cater to the central government, and the current emphasis on the HQD idea leads the local government to a breakthrough in the field of core technology.

**Table 9.** Heterogeneity test of green technology category.

| | Alternative Energy Production | Transportation | Energy Conservation | Waste Management | Administrative, Regulatory or Design Aspects |
|---|---|---|---|---|---|
| | (1) | (2) | (3) | (4) | (5) |
| Co-ordination | −0.239 | 0.790 *** | 1.500 *** | −0.624 | 0.882 ** |
| | (0.409) | (0.235) | (0.425) | (0.430) | (0.332) |
| Control Variables | Yes | Yes | Yes | Yes | Yes |
| Year | Yes | Yes | Yes | Yes | Yes |
| Region | Yes | Yes | Yes | Yes | Yes |
| Year-Region | Yes | Yes | Yes | Yes | Yes |
| Constant | −4.322 ** | −1.927 *** | −4.818 *** | −4.484 ** | −2.900 |
| | (2.014) | (0.679) | (1.318) | (1.923) | (1.942) |
| N | 21,548 | 21,548 | 21,548 | 21,548 | 21,548 |
| Adj. R$^2$ | 0.007 | 0.011 | 0.021 | 0.004 | 0.025 |

Note: figures in () are standard error; *** and ** indicate significance at the 1% and 5% levels, respectively.

### 4.4.2. Heterogeneity of Firm Ownership

In the Chinese context, firm ownership is an important institutional factor because firms with different ownership structures have vastly different cognitive logics, institutional logics, and resource endowments, which may lead to differences in firm behavior due to external regional constraints. Model 1 and Model 2 in Table 10 report the regression results of green technology choices between SOEs and private firms for regional HQCD. It is clear that only private firms respond to the green technology choice effect of regional HQCD, while SOEs may be insensitive to external constraints due to low market dynamics, solidified production and business models, and other political functions.

**Table 10.** Heterogeneity test of enterprise ownership and urban administrative level.

| | State-Owned Enterprises | Private Enterprises | High Administrative Level City | Low Administrative Level City |
|---|---|---|---|---|
| | (1) | (2) | (3) | (4) |
| Co-ordination | −0.544 | 4.198 *** | 6.314 ** | 0.488 |
| | (3.119) | (0.826) | (2.952) | (0.661) |
| Control Variables | Yes | Yes | Yes | Yes |
| Year | Yes | Yes | Yes | Yes |
| Region | Yes | Yes | Yes | Yes |
| Year-Region | Yes | Yes | Yes | Yes |
| Constant | −29.902 ** | −12.935 ** | −22.235 ** | −16.438 ** |
| | (13.673) | (5.701) | (10.377) | (7.556) |
| N | 9023 | 11,272 | 10,727 | 10,830 |
| Adj. R$^2$ | 0.027 | 0.017 | 0.039 | 0.013 |

Note: figures in () are standard error; *** and ** indicate significance at the 1% and 5% levels, respectively.

### 4.4.3. Heterogeneity of Urban Administrative Hierarchy

In China, cities with a high administrative rank enjoy many benefits, in terms such as preferential policies, financial allocations, and priority access to scarce resources, and they also assume more political functions compared to cities with a low administrative rank. In this section, provincial cities, sub-provincial cities and provincial capitals are defined as high-administrative rank cities, while prefecture-level cities are called low-administrative rank cities and are examined separately. Model 3 and Model 4 in Table 10 present the regression results, which show that HQCD in high-administrative rank cities significantly

promotes firms' green technology choices, while low-administrative rank cities do not have such incentives for firms. This can be explained by the fact that high-administrative level cities in China are the first-level responders to policies implemented by the central government and have high policy sensitivity as well as determination and the ability to implement the policies, while firms in high-administrative level cities are constrained by the local government and will make changes to accommodate the policies.

## 5. Conclusions and Policy Recommendations

### 5.1. Conclusions

With the gradual promotion of China's HQD strategy, in contrast to the majority of the literature studies on the pre-factors affecting HQD [8–11,24–31], this study focuses on the posterior influence of regional HQD, examines the impact of regional HQCD on firms' green technology choices, and further discusses its internal mechanism. The results of the study are as follows.

(1) Regional HQCD significantly promotes enterprises' green technology choices, but HQCD may need to consider that there are too many interfering factors in the implementation process [85], resulting in regional HQCD not substantially changing the direction of enterprises' green technology progress. Zhou and Yang [86] argue that coordinated regional economic–ecological development is more closely related to green innovation. After empirical analysis, we found that regional economic–ecological co-ordination not only has a green technology screening effect, but also promotes the progress of enterprises towards green practices. In further analysis, as the level of urban co-ordination increases, the intensity of its effect on the selection of green technologies and the change in the direction of green progress of enterprises gradually increases, which means that co-ordinated urban development has a self-reinforcing effect on the preference of green technology selection.

(2) After verifying that regional HQCD promotes enterprises' green technology choices, this paper further explores the mechanism of action. The results show that in the process of regional HQCD, local governments continuously optimize the market financing environment and enterprises cater to the government policy guidance, which alleviates the financial constraints of enterprises and improves corporate social responsibility, while stronger corporate financial constraints significantly inhibit green technology selection [87] and good corporate social responsibility significantly promotes the green technology screening effect [45]. In other words, regional HQCD enhances the corporate green technology screening effect by alleviating the corporate financing environment and improving corporate social responsibility.

(3) The green technology screening effect of regional HQCD is heterogeneous. Through patent category heterogeneity, regional HQCD mainly promotes enterprises' research and development of green technologies in transportation, energy saving, and administrative regulation and design; through enterprise ownership, only private enterprises respond to the green technology selection effect of regional HQCD, while SOEs may be insensitive to external constraints due to the solidified production and business model and the assumption of necessary political functions [88]; at the city administrative level, high-administrative level cities significantly promote green technology choice, while low-administrative level cities have no such incentives for firms.

### 5.2. Policy Recommendations

At present, the level of HQCD in Chinese regions is low, and the differences between regions are also large. Although HQCD promotes enterprises' green technology choices, which to some extent balances economic and environmental benefits, it does not essentially change the direction of enterprises' green technology progress, and these research findings have rich policy connotations and insights.

(1) Regional HQD does not happen overnight, and it takes a certain period of time to upgrade the industrial structure and transform the economic development mode. On

the basis of an in-depth grasp of the connotation of HQD, relevant policies should be formulated to solve existing and historical problems, achieve leapfrog development and enhance the level of regional co-ordination. Specifically, regions should implement the concept of co-ordinated development; promote integrated urban and rural development, industrial restructuring and harmonious economic and social development; adhere to green and sustainable development, reduce pollution, lower energy consumption and protect the environment; pay continuous attention to the well-being of people's livelihood, consolidate and expand the results of poverty eradication, promote the sharing of economic achievements, and enhance the level of public services and social security capabilities.

(2) The effect of green technology screening for regional HQCD should be continuously expanded. The research and development, use and promotion of green technology are important tools for sustainable development that take into account economic, social, and environmental benefits, both for local enterprises and governments. In the process of promoting high-quality construction, local governments should pay more attention to the interaction and co-ordination of the economic system and the ecosystem, and actively promote technological progress in green practices. At the same time, they should continuously optimize the market's financial environment, introduce policies and regulations, require banks and other credit institutions to simplify the process and reduce cumbersome financing costs in their financial dealings with enterprises, and support SMEs to engage in green innovative R&D activities. In addition, enterprises themselves should actively respond to the call of the local government, maintain a positive interaction with the government, enhance the awareness of corporate social responsibility, and serve the strategic needs of the government for high-quality construction with green innovation development.

(3) Formulate differentiated development strategies and take the path of HQD in special regions. As far as the regions are concerned, the central government should fully mobilize the governments at all levels when co-ordinating HQD strategies and set up a strict review mechanism. However, in the process of concrete implementation, it should fully consider the fact that cities have huge disparities in the economic base, resource endowment, and ecological environment, return the right of specific policy formulation to local governments, and formulate differentiated development strategies according to local conditions. In addition, the reform of SOEs should be continuously expanded to alleviate the state of SOE function overload and stimulate the market vitality of SOEs, in order to actively promote the transformation of SOEs to green technology in the context of HQD strategy.

*5.3. Research Deficiencies and Prospects*

The remaining limitations of this study are as follows. Firstly, the China Urban Statistical Yearbook has been updated to 2020, but its statistical indicators do not overlap with those before 2019, meaning that the regional HQD indicator system constructed in this paper can only be traced back to 2010–2019, which cannot reflect the latest reality. Subsequent studies can attempt to see if the latest relevant indicators can be obtained through the provincial level or other means, and then extend the study period. Secondly, because of the complete and easy access to the green patent information of listed companies, this paper takes listed companies as the research object, but listed companies only account for a small portion of all enterprises in the region, which may have some influence on the research conclusions. Subsequent studies can use the data of Chinese industrial enterprises to match and screen the patent information published by the State Intellectual Property Office of China to expand the sample size and increase the universality of the conclusions. Finally, in performing the mechanism analysis, this paper only found two channels of corporate financing constraints and corporate social responsibility awareness, but there may be more inherent mechanisms for subsequent research.

**Author Contributions:** Conceptualization, software, visualization and writing—original draft, D.H.; project administration, C.Z.; validation, H.G.; funding acquisition, C.L.; formal analysis, methodology, D.H. and C.Z.; data curation, D.H. and H.G.; investigation, H.G. and Y.Z.; resources, C.Z., Y.Z. and C.L.; supervision, C.Z. and C.L.; writing—review and editing, D.H., Y.Z. and C.L. All authors have read and agreed to the published version of the manuscript.

**Funding:** This research was funded by Zhejiang Province Philosophy and Social Science planning annual topic (NO. 22NDJC072YB), Zhejiang Province Soft Science Research Project (No. 2021C35087), the National Natural Science Foundation of China (No. 71704087), the National Natural Science Foundation of China (No. 71673182), Introduction project of Yunnan University of Finance and Economics (No. 20190043).

**Institutional Review Board Statement:** Not applicable.

**Informed Consent Statement:** Not applicable.

**Data Availability Statement:** The data presented in this study are available on request from the corresponding author.

**Conflicts of Interest:** The authors declare no conflict of interest.

## Appendix A

1. Range normalization. For positive indicators: $X'_{ijt} = \frac{X_{ijt} - \min\{X_j\}}{\max\{X_j\} - \min\{X_j\}}$, for negative indicators: $X'_{ijt} = \frac{\max\{X_j\} - X_{ijt}}{\max\{X_j\} - \min\{X_j\}}$, $X'_{ijt}$ is the normalized metric, i means different cities, j represents different indicators, t means years. For writing convenience, $X_{ijt}$ is still used to represent the standardized indicator.

2. The information entropy and entropy redundancy of item j are calculated. $E_j = -1 \Big/ \ln(2838) \sum_{i=1}^{284} \sum_{t=1}^{t=10} (P_{ijt} \times \ln P_{ijt})$, $D_j = 1 - E_j$, where $P_{ijt} = \frac{X_{ijt}}{\sum_{i=1}^{284} \sum_{t=1}^{10} X_{ijt}}$ is the probability of $X_{ijt}$ under the index sample.

3. Calculate the weight of the index of j and the composite index score of city i in period t. $W_j = \frac{D_j}{\sum_{j=1}^{m} D_j}$, $S_{it} = \frac{S^-_{it}}{S^+_{it} + S^-_{it}}$, where $S^+_{it} = \sqrt{\sum_{j=1}^{m} (c_{ijt} - c^+_{ijt})^2}$ represents the distance to the positive ideal solution, $S^-_{it} = \sqrt{\sum_{j=1}^{m} (c_{ijt} - c^-_{ijt})^2}$ represents the distance to the negative ideal solution. $c_{ijt}$ is the index processed by the original data weighting specification.

4. Measure the coupling between the five systems. $C_{it} = \frac{\sqrt[5]{S^A_{it} \times S^B_{it} \times S^C_{it} \times S^D_{it} \times S^E_{it}}}{(S^A_{it} + S^B_{it} + S^C_{it} + S^D_{it} + S^E_{it})/5}$.

5. Aggregate the combined scores of the five systems. $T_{it} = \alpha_1 S^A_{it} + \alpha_2 S^B_{it} + \alpha_3 S^C_{it} + \alpha_4 S^D_{it} + \alpha_5 S^E_{it}$. Considering that the five systems are equally important for the HQD of cities, the undetermined coefficients are all set as 0.2.

6. Measure the co-ordination of the five systems. $Coordination_{it} = \sqrt{C_{it} \times T_{it}}$.

## Appendix B

Figure A1 shows the trend of HQCD of 284 cities in China from 2010 to 2019. Although the average co-ordination degree showed a slow rise from 2010 to 2019, the co-ordination level in 10 years was in six stages from moderate imbalance to moderate co-ordination, and the number of cities with high co-ordination level was significantly less than that of cities with low co-ordination level. This shows that the overall progress of urban HQCD in China is slow, the level is low and the differences between cities are huge. In order to further explore the current situation of HQCD in China, this paper draws the geographical location map of the degree of HQCD in China in 2019 (see Figure A2). It can be observed that most cities in China are still in two stages of mild imbalance and near imbalance, and their co-ordination level is relatively low. A few cities with better co-ordinated development are

mainly concentrated in southeast coastal areas, and a few are distributed in central north China and part of northwest China.

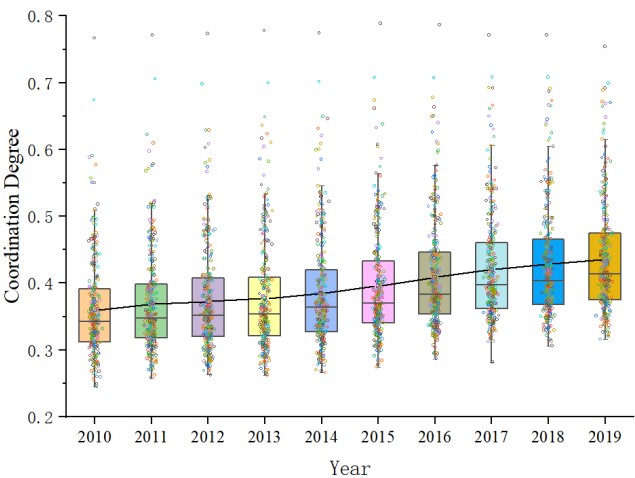

**Figure A1.** The trend of HQCD in 284 cities.

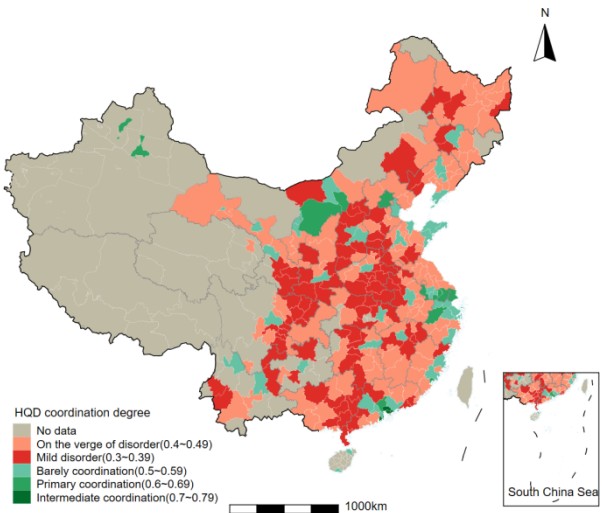

**Figure A2.** Geographical distribution of HQCD in 2019.

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
