# Peer review of "How Regional High-Quality Co-Ordinated Development Influences Green Technology Choices: Evidence from 284 Cities in China"

_land, doi:10.3390/land11071111_

Round 1

Reviewer 1 Report

Attached

Reviewer 2 Report

The manuscript titled " How Regional High-Quality Coordinated Development Influences Green Technology Choices: Evidence from 284 Cities in China " intends to answer the follow question: Can the high-quality coordinated development (HQCD) strategy implemented by local governments influence micro-enterprises, to guide and screen enterprises' green technology choices, and what are the hidden mechanisms? The manuscript select data from 284 cities in China from 2010 to 2019 by using entropy weight TOPSIS method and coupling coordination degree model.

The research is original; it could be characterized as novel and in my opinion important to the field, it also has an almost appropriate structure, and the language has been used well. In the meanwhile, the manuscript has a big extent (about 9,100 words) and it is comprehensive. The tables (10) and figures (2) make the paper reflect well to the reader. For this reason, paper has a "diversity look", not only tables, not only numbers, not only words. It is advised to revise tables, compare them, or use an appendix if you agree.

Please reduce the manuscript (about 20%), you must follow the instructions of the Journal [see: Instructions for Authors / Manuscript Submission Overview / Types of Publications - (https://www.mdpi.com/journal/land/instructions#submission)]. Using an appendix probably helps you.

The title, I think, is all right. The abstract did not reflect well the findings of this study, and it has not the appropriate length. Please revise the abstract of the manuscript and do not forget abstract need to encourage readers to download the paper. The Abstract needs further work. It is not clear. Abstracts should indicate the research problem/purpose of the research, provide some indication of the design/methodology/approach taken, the findings of the research and its originality/value in terms of its contribution to the international literature. The abstract has a long length (about 276 words). Please, revise the abstract, it must be up to 200 words long, for this reason I would be good to reduce [see: Instructions for Authors / Manuscript Submission Overview / Accepted File Formats - (https://www.mdpi.com/journal/land/instructions#submission or https://www.mdpi.com/files/word-templates/land-template.dot)].

Please, revise the manuscript and make the appropriate currency exchange using dollars ($) or euros (€) or both also you can keep yuan (line 41). This is because the results of the research must be directly comparable to other similar surveys that have already been carried out around the world and other such surveys will certainly be carried out, and do not forget, the journal “Land” is international.

The introduction is effective, clear, and well organized; it really introduced and puts into perspective what research is negotiating but is very big. Please revise the Introduction of the manuscript and include references which already exist in bibliography (as you did). For the Methodology chapter, the research conduct has been tested in several areas of the world, with similar results and will probably be tested in others. Appropriate references to the methodology included in the already published bibliography. It is advised to revise the Discussion and Conclusion. Both sections should be consistent in terms of Proposal, Problem statement, Results, and of course, future work (as you did). Your conclusion section is big and does not justice to your work. Make it your key contributions, arguments, and findings clearer. You must refer to the literature and previous studies in your discussion section.

More discussion is needed, comparing the results of this work related to attributes with those of other studies. I believe that the conclusions section or discussion should also include the main limitations of this study and incorporate possible policy implications (as you did). I think something more should be said about practical implications.

Please revise the references of the manuscript and include references which already exist in the bibliography. References must have an appropriate style, for this reason I would be good to change [see: Instructions for Authors / Manuscript Preparation / Back Matter / References: - (https://www.mdpi.com/journal/land/instructions or https://www.mdpi.com/authors/references)]. Do not forget, DOI numbers (Digital Object Identifier) are not mandatory but highly encouraged and make the review easier.

For example, for reference 5 you write “Chai, J. The Impact of Green Innovation on Export Quality. Applied Economics Letters 2022, 1–8, doi:10.1080/13504851.2022.2045249”. I think it must be revised as “Chai, J. The impact of green innovation on export quality. Appl. Econ. Lett. 2022, 1–8, doi:10.1080/13504851.2022.2045249”.
